# 19th International Symposium on Epstein–Barr Virus and Associated Diseases, 29–30 July 2021, Asahikawa, Japan

**DOI:** 10.3390/cancers14122924

**Published:** 2022-06-14

**Authors:** Takumi Kumai, Miki Takahara, Yasuaki Harabuchi

**Affiliations:** 1Department of Innovative Head and Neck Cancer Research and Treatment, Asahikawa Medical University, Asahikawa 078-8510, Japan; 2Department of Otolaryngology-Head and Neck Surgery, Asahikawa Medical University, Asahikawa 078-8510, Japan; miki@asahikawa-med.ac.jp (M.T.); hyasu@asahikawa-med.ac.jp (Y.H.); 3Department of Otolaryngology-Head and Neck Surgery, Hokuto Hospital, Obihiro 080-8033, Japan

**Keywords:** 19th international symposium on EBV, Epstein–Barr virus, gastric cancer, nasopharyngeal cancer, NK/T-cell lymphoma

## Abstract

**Simple Summary:**

Novel insights into the basic and translational findings on Epstein–Barr virus-related diseases were presented at the “19th International Symposium on Epstein–Barr Virus and Associated Diseases” in Asahikawa, Japan.

**Abstract:**

Novel insights into Epstein–Barr virus (EBV) pathogenicity were presented at the “19th International Symposium on Epstein-Barr Virus and Associated Diseases” in Asahikawa, Japan. In addition, basic and translational findings on EBV-associated tumors, including natural killer (NK)/T-cell lymphoma, gastric cancer, and nasopharyngeal cancer, were presented by an international group of scientists and clinicians.

## 1. Introduction

The “19th International Symposium on Epstein-Barr Virus and Associated Diseases” was held from 29–30 July 2021, at the Art Hotel in Asahikawa, Japan (Figure 1). It was postponed in 2020 due to the COVID-19 pandemic and later held in-person or as on-demand online streaming of presentations for two weeks from 2–15 August 2021 (Figure 2). The symposium included more than 200 participants from 14 nations, including 51 onsite and online speakers, 8 speakers at special lectures, such as the Henle lecture, and 112 poster presenters. Due to the time difference, the online presentations were allocated to morning or evening sessions according to the presenter’s geographic location. The speakers covered topics ranging from basic to translational research and novel clinical findings in Epstein–Barr virus (EBV)-related diseases. These included the pathogenic role of EBV in various malignancies, such as nasopharyngeal cancer, gastric cancer, and lymphoma, and the mutual relationship between EBV and immune cells. In this meeting report, we summarize the contents of the oral presentations given at the symposium.

## 2. Contents of Oral Presentations and Seminar Given on 29 July 2021

Session 1 focused on viral latency and pathogenesis. Jayaraju Dheekollu (The Wistar Institute) demonstrated that an amino acid residue in Epstein–Barr nuclear antigen 1 (EBNA1) is required for episomal maintenance in EBV. Tyrosine 518 is indispensable for forming an EBNA1–DNA crosslink with the EBV origin of plasmid replication *oriP* and single-strand cleavage, which is essential for replication termination and viral episomal maintenance. EBV-associated tumors suppress the expression of lytic and latent EBV antigens to evade immune detection. Rui Guo (Brigham and Women’s Hospital) demonstrated that EBV^+^ Burkitt cells (BLs) subvert the methionine and folate metabolism pathways to support the EBV genome hypermethylation necessary for viral antigen silencing. Methionine restriction derepresses EBV lytic and latency antigens in EBV-infected BL cells, disrupting type I latency. Chong Wang (Brigham and Women’s Hospital) compared resting B cell and lymphoblastoid cell line (LCL) genome-wide chromatin interaction maps to identify the genome architecture changes arising from EBV infection. The virus partly controls lymphocyte growth by globally reorganizing host genome architecture to facilitate the expression of key oncogenes. Emmanuela N Bonglack (Duke University School of Medicine) showed that the monocarboxylate transporters (MCTs) 1 and 4 responsible for lactate export and cancer cell proliferation are induced by Epstein–Barr nuclear antigen 2 (EBNA2) and latent membrane protein 1 (LMP1), respectively. Dual MTC 1/4 inhibition suppresses the growth of LCLs and sensitizes them to killing by the electron transport chain complex 1 inhibitors phenformin and metformin. Jillian Bristol (University of Wisconsin–Madison) applied bulk/single-cell RNA sequencing (RNA-Seq) and immunoblot analysis to compare cellular and viral gene expression in early-passage LCLs infected with type I or type II EBV. Enhanced lytic viral reactivation in type 2 EBV-infected B cells is associated with B cell receptor (BCR) signaling increased by interferon regulatory factor 4 (IRF4) downregulation. Huanzhou Xu (University of Florida) examined proteins at cellular replication forks, i.e., replisomes, in EBV-transformed B cells using the isolation of proteins on nascent DNA (iPOND) and mass spectrometry. Eight novel proteins are associated with replisomes at cellular replication forks in LCLs and BLs. One zinc finger protein acts directly on fork progression and enhances B-cell proliferation.

Session 2 focused on viral replication and reactivation. Eric M Burton (University of Florida) reported that the NOD-, LRR-, and pyrin domain-containing protein 3 (NLRP3) inflammasome, a cellular defense system against infectious agents, initiates an EBV lytic cascade in response to danger signals. Entry into the lytic phase allows the virus to escape from damaged cells. Nick Van Sciver (University of Wisconsin–Madison) demonstrated that the Hippo signaling effectors, yes-associated protein 1 (YAP), and tafazzin (TAZ) induce the EBV lytic cycle in epithelial cells. Their known activator lysophosphatidic acid, commonly expressed in saliva, promotes EBV lytic reactivation in epithelial cells via a YAP/TAZ-dependent mechanism. Quincy Rosemarie (University of Wisconsin–Madison) used a library of 293 cells carrying EBV with single-gene knockouts of its lytic replication genes. The reorganization of chromatin in the EBV lytic phase is required for expressing core lytic replication genes but not late lytic genes. Dinesh Verma (University of Utah School of Medicine) revealed that EBV lytic reactivation from latency enhances severe acute respiratory syndrome-coronavirus 2 (SARS-CoV-2) entry into epithelial cells. The EBV Zta protein, essential for EBV lytic cycle entry, upregulates cellular angiotensin-converting enzyme-2 (ACE2), a SARS-CoV-2 receptor, during EBV lytic replication. Although anti-ACE2 antibodies block the ACE2-dependent entry of SARS-CoV-2 into epithelial cells, EBV infection enhances it. Rodney P Kincaid (Oregon Health and Science University) used fluorescence-activated cell sorting-based genome-wide CRISPR/Cas9 knockout screening and single-cell RNA sequencing. Viral- and host-encoded miRNAs modulate the EBV transcriptional reactivation process at the single-cell level.

At the luncheon seminar, Yoshihisa Yamano (St. Marianna University School of Medicine) presented the pathogenesis and genomic changes during leukemic transformation in patients with human T-lymphotropic virus type 1 (HTLV-1)-associated neuroinflammatory diseases. Patients with HTLV-1 can develop aggressive adult T-cell leukemia/lymphoma (ATLL) and/or the debilitating inflammatory disease HTLV-1-associated myelopathy/tropical spastic paraparesis (HAM/TSP). Yamano demonstrated that HTLV-1 induces a C-X-C motif chemokine receptor 3-positive (CXCR3^+^) Th1-like state in infected CD4^+^CCR4^+^ T cells via T-box transcription factor expression. The cells consequently produce interferon (IFN)-γ to stimulate astrocytes. Stimulated astrocytes secrete CXCL10, which recruits infected cells via CXCR3, constituting a T-helper type 1-centric positive feedback loop. This loop causes chronic inflammation in the central nervous system in HAM/TSP. Yamano also showed the genomic changes predicting the features that lead to the development of ATLL in patients with HAM/TSP.

Session 3 focused on nasopharyngeal cancer (NPC) and gastric cancer. Satoru Kondo (Kanazawa University) showed that Japanese patients with NPC possess type 2 EBV or EBV strain associated with intermediate risk (BALF2 H-H-L), which is different from that in the endemic region of Southern China (BALF2 H-H-H). Xia Yu (Zhongshan Hospital of Sun Yat-sen University) reported that the diagnostic yield for NPC can be improved by measuring an anti-BNLF2b antibody in a study of more than 30,000 patients with NPC. Anti-BNLF2b antibody has better specificity and sensitivity as a diagnostic marker than EBV-DNA, anti-VCA IgA, or anti-EBNA1 IgA. Qian Zhong (Sun Yat-sen University Cancer Center) performed the single-cell RNA sequencing of NPC biopsy samples. Malignant cells expressing dual immune and epithelial features are associated with poor prognosis. This dual feature positively correlates with the expression of co-inhibitory receptors on CD8^+^ tumor-infiltrating T cells. Atsushi Okabe and Harue Mizokami (Chiba University) analyzed the genome-wide chromatin topology of normal and gastric cancer cells obtained by chromosome conformation capture combined with high-throughput sequencing. The EBV genome directly interacts with and remodels the host heterochromatin via epigenetic modifications, facilitating proto-oncogene expression and tumorigenic transformation. This phenomenon, termed “enhancer infestation,” can be observed in EBV^+^ gastric and nasopharyngeal carcinoma cell lines. Mariko Yasui (The University of Tokyo) analyzed the gene expression of monolayer and sphere-forming cells from EBV^+^ gastric carcinoma (EBVaGC). The latent membrane protein 2A (LMP2A) contributes to maintaining sphere-forming cancer stem-like cells through the nuclear factor-kappa B (NF-κB) pathway in EBVaGC. Soichiro Fukuda (Yamaguchi University Graduate School of Medicine) proved that MC180295, a novel demethylating compound, could decrease the expression of BRCA1 and cell cycle-related genes and inhibit the proliferation of EBVaGC cell lines by suppressing the DNA and cell cycle. Hisashi Iizasa (Shimane University) reported that *Helicobacter pylori* increases the levels of low-affinity EBV receptors ephrin type-A receptor 2 (EphA2) and non-muscle myosin IIA (NMHC-IIA) and elevates the efficiency of EBV infection in gastric epithelial cells. This elevation depends on the adhesion of *H. pylori* to epithelial cells.

In an invited lecture, Kwok-Wai Lo (The Chinese University of Hong Kong) presented the whole-genome profiling of EBV-associated nasopharyngeal carcinoma. It is the most extensive whole-genome catalog of EBV-associated NPC and viral oncogene expression, revealing the mutational pattern of the genome, including a new homologous recombination defect signature prevalent in NPC. Moreover, recurrent structural alterations and homozygous deletions alter NF-κB and transforming growth factor (TGF)-β signaling, cell cycle, and impaired immunity. These aberrations and viral gene expression undermine innate and adaptive immunity in 79% and 53% of the cases, respectively, indicating that viral and somatic events subvert antitumor immunity in most patients with NPC. In addition, the catalog showed frequent alterations in TGF-β receptor type 2 TGFBR2 and cyclin-dependent kinase inhibitor 2A (CDKN2A), which define a novel viral–somatic interaction promoting persistent EBV infection for NPC pathogenesis.

Mei-Ru Chen (National Taiwan University) delivered keynote lecture 1, describing how EBV overcomes cellular barriers and reorganizes cellular organelles for virion replication and assembly. Various EBV lytic proteins regulate and modulate the nuclear envelope structure, such as viral BGLF4 kinase and the nuclear egress complex BFRF1/BFRF2. After nuclear egress, lesser amounts of viral BGLF4 kinase and the single-stranded DNA-binding protein BALF2 are present in the concave nuclear region of TW01-EBV cells, along with tegument proteins. Thus, EBV appears to use multiple types of cellular machineries for maturation, which may disturb cellular homeostasis and lead to EBV pathogenesis.

Session 4 focused on diseases and disorders, including chronic active EBV infection (CAEBV), EBV-associated T/NK cell lymphoproliferative diseases (EBV-T/NK-LPD), nasal NK/T-cell lymphoma, and post-transplant lymphoproliferative disorders (PTLDs). CAEBV is a progressive disease with persistent inflammatory symptoms accompanied by clonally proliferating EBV^+^ T or NK cells in the peripheral blood (EBV-T/NK-LPD). Ayaka Ohashi (St. Marianna University School of Medicine) showed that the plasma level of interleukin (IL)-1β in CAEBV could be a biomarker of angiopathy. Peripheral blood monocytes are considered a source of IL-1β. Yuriko Ishikawa (National Center for Child Health and Development) elucidated the role of mucosal-associated invariant T cells in EBV-T/NK-LPD by checking peripheral blood from patients. The activation level of the cells depends on disease severity and IL-18 levels; thus, they may be involved in EBV T/NK-LPD immunopathogenesis. Paul J Collins (University of Birmingham) reported that an increase in myeloid-suppressor cells supports immune evasion, leading to EBV^+^ T/NK cell proliferation in patients with CAEBV. Conventional steroid therapy for symptom relief might exacerbate the underlying disease. Hydroa vacciniforme-like lymphoproliferative disorder (HV-LPD) is a cutaneous form of CAEBV. Keiji Iwatsuki (Okayama University Graduate School of Medicine) observed atypical αβT and γδT cells expressing NK cell markers (CD16 and CD56) in HV-LPD. Miki Takahara (Asahikawa Medical University) presented the clinical characteristics of 62 patients with early-stage nasal NK/T-cell lymphoma. The five year disease-specific survival rate of the 18 patients treated with concurrent arterial infusion chemoradiotherapy was 94%. Mayumi Yoshimori (St. Marianna University School of Medicine) demonstrated that EBV^+^ NK cells produce IFN-γ, enhancing the differentiation and activation of macrophages. The activation leads to lethal complications such as hemophagocytic lymphohistiocytosis. Nenad Sejic (Walter and Eliza Hall Institute for Medical Research) reported that B-cell lymphoma 2 (BCL2) homology domain 3 (BH3)-mimetic drugs, small compounds that antagonize anti-apoptotic BCL-2 family proteins, decrease BCL-XL expression, inducing the apoptosis of CAEBV and NK/T-cell lymphoma cell lines. Julien Lupo (Grenoble Alpes University Hospital) detected soluble ZEBRA proteins (sZEBRA) with an ELISA assay in plasma samples from more than 300 patients who underwent organ transplantation. The protein levels were higher in patients with PTLD and graft-versus-host disease. Measuring sZEBRA proteins could help identify patients likely to develop severe outcomes during the critical post-transplant period.

Paul Farrell (Imperial College London) presented the course of his EBV research history at the Henle Lecture. He first discussed the nature of EBV, which is associated with a wide range of diseases, including infectious mononucleosis, and contributes to several types of cancer, but is also possibly involved in other chronic diseases. A comprehensive understanding of the EBV genome has paved the way for modern EBV research, establishing coherent models for diverse types of EBV infections. The EBV genome variations, type 1 and 2, induce phenotypic changes, and some geographic variations may explain the diverse incidence of several EBV-related diseases. Moreover, some deletions have been observed in the EBV genome infected with T or NK cells and can serve as markers for the disease. In addition, drugs that cause EBV reactivation can be used in patients with EBV^+^ gastric carcinoma.

## 3. Contents of Oral Presentations and Seminar Given on 30 July 2021

Ayako Arai (St. Marianna University School of Medicine) presented recent discoveries on CAEBV in a morning seminar. Ever since CAEBV was added to the World Health Organization classification of tumors of hematopoietic and lymphoid tissues in 2017, numerous reports on the disease have been recorded. Retrospective surveys in Japan have shown that chemotherapies are insufficient to resolve disease activity and eradicate infected cells. Because the only option for curing CAEBV is allogeneic hematopoietic stem cell transplantation, establishing novel treatments is needed. Constitutively activated NF-κB and STAT3 pathways contribute to immortalization and cytokine production in EBV^+^ CAEBV cells. Hence, these pathways may be attractive therapeutic targets for eradicating these cells in CAEBV.

Session 5 focused on virus–host interactions and immunity. Elliott SoRelle (Duke University) profiled the transcriptomes of five different EBV-infected LCLs with single-cell RNA-Seq analysis, revealing a considerably heterogenic LCL transcriptome profile. These findings indicate that LCL is a continuum of various transcriptional states. Post-infection dynamic studies will provide a complete picture of EBV-induced events. Anna Gil (University of Massachusetts Medical School) reported that CD4^+^/CD8^+^ T cells and the ratio of CD4/CD8 T cells are upregulated in the peripheral blood mononuclear cells (PBMCs) of patients with multiple sclerosis (MS) and correlate with disease severity and the alteration of immune response to EBV. These results suggest that CD4^+^CD8^+^ T cells play a pathogenic role in multiple sclerosis. Pascal Polepole (University of Nebraska) found that xenografts of LCL exacerbate experimental autoimmune encephalomyelitis, which mimics a key feature of MS, in C57BL/6 mice, suggesting that EBV-induced B-cell proliferation exacerbates MS. These xenografts have diverse changes in the gut microbiome and modulate the blood–brain barrier. Adityarup Chakravorty (University of Wisconsin–Madison) demonstrated that the EBV oncogene EBNA1 directly suppresses NK cell activation by repressing the NK cell-activating NKG2D ligands ULBP1 and ULBP5 in infected B cells. Because EBNA1 is expressed in all EBV-related tumors, its suppression may inhibit tumor growth. Ashely Campbell (University of Toronto) performed affinity purification coupled with mass spectroscopy, identifying host proteins interacting with EBV protein BGLF2. They interact with the miRNA-induced silencing complex (RISC) and interfere with cellular miRNA function. Moreover, BGLF2 inhibits let-7 miRNA and induces SUMOylation by increasing free SUMO levels. Phillip Ziegler (University of Pittsburgh Medical Center) proved that pseudostratified epithelial cells are susceptible to EBV infection using 3D air–liquid interface cultures. Moreover, all four cell types, basal, suprabasal, mucosecretory, and ciliated, can harbor the infection. Single-cell RNA profiling of the cells identified many EBV transcripts associated with latent, abortive, and lytic infections.

Session 6 focused on lymphomagenesis and therapeutics. Ben Gewurz (Harvard Medical School) showed that ferrireductase CYB561A3 is critical for the proliferation of Burkitt’s lymphoma, but not LCLs. CYB561A3 knockout Burkitt lymphoma cells show catastrophic lysosomal and mitochondrial damage and impaired mitochondrial respiration. Wei Bu (National Institutes of Health) identified a set of human monoclonal antibodies targeting five different antigenic sites on EBV gH/gL. These antibodies prevent EBV infection in both epithelial and B cells. One of the antibodies causes nearly complete protection of humanized mice from viremia after the EBV challenge. Ibukun A Akinyemi (University of Florida) found that the c-Fos/activator protein (AP)-1 inhibitor T-5224 inhibits BamHI Z fragment leftward open reading frame 1 (BZLF1) promoter activity, viral lytic gene expression, and EBV production in Burkitt lymphoma cells. It also retards the immortalization and outgrowth of LCLs, possibly via ZEBRA/phosphoinositide 3-kinase (PI3K) pathway inhibition. Blachy J Davila Saldana (George Washington University) assessed an international cohort of 59 patients with CAEBV outside Asia, showing that hematopoietic stem cell transplantation (HSCT), especially early in the disease course, is the only curative therapy for CAEBV. Sang-Hoon Sin (University of North Carolina at Chapel Hill) generated a transgenic mouse (d.197) carrying the entire 140,000 bp genome of Kaposi’s sarcoma-associated herpesvirus. Within 100 days, 10% of the mice developed angiosarcoma, which mimicked human Kaposi’s sarcoma pathology. Sandhya Sharma (Baylor College of Medicine) successfully induced functional T cells reactive to four EBV type 2 latency proteins from CD45RA-depleted PBMCs. These T cells decrease the metastatic spread of autologous EBV^+^ tumor cells in a murine xenograft model.

At the luncheon seminar, Hideaki Nakajima (Yokohama City University Graduate School of Medicine) described the role of *O*-GlcNAcylation and the addition of *O*-linked β-N-acetyl-d-glucosamine moiety to serine or threonine residues in hematopoiesis. This moiety is conserved throughout eukaryotes and is a fundamental regulator of numerous biological processes. For instance, *O*-GlcNAcylation has essential and pleiotropic roles in the differentiation, proliferation, and function of various cells, such as T, B, and polyphyletic progenitor cells. Aberrant *O*-GlcNAcylation promotes hematological malignancies through disordered epigenetics, metabolism, and gene transcription.

Session 7 focused on viral infections and immunity. Takayuki Murata (Fujita Health University School of Medicine) showed that EBV infection induces programmed death-ligand 1 (PD-L1) expression in primary B cells through EBNA2, which binds to the promoter region of the PD-L1 gene. Accordingly, EBV-infected B cells may escape antiviral immunity via the PD-1/PD-L1 pathway. Keiko Nagata (Tottori University) demonstrated that the reactivation of EBV in B cells produces autoreactive antibodies, which may exacerbate Graves’ disease. Danling Dai (Sun Yat-sen University Cancer Center) reported that the EBV transcriptional factor BZLF1 decreases the N6-methyladenosine modification of Kruppel-like factor 4 (KLF4), subsequently increasing its protein level. The KLF4 upregulation promotes EBV infection of nasopharyngeal epithelial cells, resulting in a positive feedback loop between EBV and host molecules. Yoshitaka Sato (Nagoya University Graduate School of Medicine) showed that EBV-infected cells release exosomes containing EBV tegument protein BGLF2. Exosome-encapsulated BGLF2 enhances EBV gene expression, supporting the viral infection. These results suggest that tegument proteins are functional in extra-viral particles, such as exosomes. Xiangwei Kong (Sun Yat-sen University Cancer Center) created a unique model of a varicella-zoster virus (VZV)-based EBV vaccine, displaying the prefusion form of EBV glycoprotein B on its surface. Vaccination against EBV glycoprotein B elicits an EBV neutralization effect in both B and epithelial cells.

EBV is associated with 10% of gastric carcinomas (EBVaGCs). Unique clinicopathological features have been indicated in EBVaGC, such as male predominance, predominant location in the proximal stomach, lymphoepithelioma-like histology, and a favorable prognosis. Masashi Fukayama (Asahi General Hospital) presented his findings on EBVaGC in an invited lecture. In the earliest phase of EBVaGC development, single or a few EBV-infected epithelial cells initiate neoplastic proliferation within the stomach mucosa, potentially via epigenetic regulation such as DNA methylation. EBVaGC cells harbor characteristic genetic alterations, including mutations in AT-rich interaction domain 1A (ARID1A) and PD-L1. ARID1A protein deficiency increases susceptibility to EBV, and PD-L1 overexpression allows the tumor cells to escape the host immune system. Concerning EBV contribution, LMP2A maintains cancer stem-like cells and resistance to apoptosis. Moreover, EBV-derived microRNAs participate in EBVaGC transformation and remodeling of the tumor microenvironment by affecting surrounding stromal cells through exosomes.

EBV-associated T/NK-cell malignancies are refractory to conventional chemotherapy and have poor prognoses. Although the transforming capacity of EBV in B cells is well known, the molecular pathogenesis of EBV-caused clonal expansion in T/NK cells is still elusive. As the keynote speaker 2, Hiroshi Kimura (Nagoya University Graduate School of Medicine) described the relationships between EBV and T/NK cells and possible infection of progenitor cells prior to T- or NK-cell differentiation, suggesting a new explanation for how EBV infects T/NK cells. Moreover, EBV gene deletions around the BART microRNA cluster region have been found in CAEBV and EBV-associated T/NK-cell lymphoma, indicating that these deletions are responsible for the neoplastic proliferation of EBV-infected cells.

Session 8 focused on viral variation and the environment. Misako Yajima (Tohoku Medical and Pharmaceutical University) presented an approach to clone and sequence the entire EBV genome obtained from tonsil-derived spontaneous LCLs, uncovering seven distinct EBV strains. Phylogenetic analyses revealed that the subgroup of nasopharyngeal carcinoma-derived EBV strains in endemic areas differs from that in non-endemic areas, such as Japan. Yusuke Okuno (Nagoya University Hospital) found 150 microdeletions in the EBV genome derived from various EBV-related malignancies and diseases. Because most of the deleted lesions are shared between the disorders, specific deletions may contribute to the neoplastic phenotype of infected cells. Xiao Zhang (Sun Yat-sen University Cancer Center) presented the ultrastructure of EBV virion components—the icosahedral capsid, dodecameric portal, and capsid-associated tegument complex—using cryo-electron microscopy. This model helps understand the mechanisms of EBV genome retention and ejection. Yufeng Chen (Karolinska Institute) reported that cigarette smoking is associated with EBV reactivation, as defined by serum IgA antibody levels against EBV capsid antigen and EBNA1. Other environmental factors, such as alcohol and salted fish consumption and various residential and occupational exposures, were not associated with EBV reactivation. Ezgi Akidil (German Research Center for Environmental Health) succeeded in genome engineering primary resting human B cells using CRISPR-Cas9 technology. In this system, B cells carry a knockout of CDKN2A, which encodes the cell cycle regulator p16^INK4a^. Infecting B cells with EBNA3 mutant EBV strain showed that EBNA3C controls p16^INK4a^. Vishwanath Kumble Bhat (University of Sussex) identified novel variations in EBV EBNA1 sequences from specific geographical regions. Some variations cause amino acid changes in the EBNA1 DNA-binding domain, influencing DNA binding affinity. As EBV manipulates autophagy during the latent and lytic cycles, Maria Pena Francesch (University of Zurich) discovered that the capsid scaffold proteins BVRF2 and BdRF1 interact with the autophagosome marker LC3B. This interaction affects virus production through the virus assembly process.

In addition to the oral presentations in the main symposium, 112 posters presented online. Ka Wo (The University of Hong Kong) showed that most infectious mono-derived EBV genomes belong to type 1 EBV. A genome-wide association study did not detect any EBV variant that is specific to EBV mono-infection. Yoshitaka Aoki (Kanazawa University) showed the prevalence of EBV in the adenoid and palatine tonsils of adults. Viral DNA is detectable in 74% and 71% of the adenoids and tonsils, respectively. Asuka Nanbo (Nagasaki University) described the mechanism of releasing EBV virions from the host cell into the extracellular milieu. Three Rab GTPases, Rab8a, Rab10, and Rab11a, induce the release of EBV infectious virions via the secretory pathway.

Julia Myers (LSUHSC–Shreveport) described a mechanism of viral interference. Because HPV16 E7 facilitates the degradation of retinoblastoma protein (Rb) to promote S-phase progression, Rb knockdown rafts exhibit reduced EBV replication. Xiaomei Deng (Sun Yat-sen University) demonstrated that dengue virus infection reactivates EBV through the PI3K pathway. Likewise, EBV infection promotes dengue virus propagation in EBV-infected cells, suggesting that some viruses can coexist with EBV.

p62 is a ubiquitin sensor and signal-transducing adaptor with multiple functions in diverse contexts. Shunbin Ning (East Tennessee State University) showed that p62 participates in LMP1 signal transduction via TRAF6 ubiquitination and activation. Silencing p62 impairs the ability of LMP1 to regulate target gene expression. Ling Wang (East Tennessee State University) reported that p62 depletion causes a substantial activation of DNA repair mechanisms. Eiji Kobayashi (Kanazawa University) described how LMP1 is sorted into exosomes. LMP1 is physically associated with the ubiquitin C-terminal hydrolase L1 (UCH-L1), and the C-terminal farnesylation of UCH-L1 directs LMP1 to exosomes. Inhibiting farnesylation reduces LMP1 amounts in exosome fractions. Thus, restricting the exosome-mediated transfer of prometastatic molecules during cell-to-cell communication is a potential therapeutic target.

Blue-leaf A Cordes (University of Wisconsin) showed that hypoxia induces the lytic cycle through BZLF1. Hypoxia-inducing reagents kill EBV^+^ gastric cancer cells. Lok Yiu Avala Ngan (The University of Hong Kong) demonstrated that only early passages of nasopharyngeal cancer cells are activated to undergo lytic infection by hypoxia, suggesting that passage number is critical in research using nasopharyngeal cancer cell lines. Hyemi Kim (Yonsei University College of Medicine) showed that gemcitabine induces lytic activation in EBV-associated cancers. The ZID domain plays a significant role in the lytic cycle entry through p53, which was confirmed in mouse- and patient-derived cancer organoid models. By contrast, Jun Li (Qingdao University) showed that the EBV latent protein LMP1 induces p53 protein expression via the lncRNA H19/miR-675-5p axis. Thus, the function of p53 in the balance between EBV lytic and latent cycles may be context-dependent. The latent EBV protein is considered to drive oncogenesis. However, Hirotomo Dochi (Kanazawa University) demonstrated that ZEBRA, a trigger regulator of the lytic cycle, is expressed in EBV^+^ nasopharyngeal cancer and is related to a worse prognosis.

Immunoregulation may be critical to developing curative treatments for virus-related diseases. Shouhei Miyagi (Nagoya University Graduate School of Medicine) reported the inhibition of stimulator of interferon genes (STING), a central adaptor that allows DNA sensors to recognize exogenous cytosolic DNA to activate the type I interferon signaling cascade and reduces the transformation activity of EBV. Sho Sasaki (Yamaguchi University Graduate School of Medicine) and Kina Kase (Kanazawa University) demonstrated in separate studies that EBV^+^ gastric and nasopharyngeal cancers express PD-L1, a negative immune checkpoint. Yui Hirata-Nozaki (Asahikawa Medical University) found that the c-Met protein contains a helper CD4 epitope that elicits antitumor T cell responses against EBV^+^ NK/T-cell lymphoma. In addition, HGF-c-Met signaling induces NK/T-cell lymphoma proliferation in an autocrine manner. Sandhya Sharma (Texas Children’s Hospital) succeeded in inducing EBV-specific T cells that react to lytic and latent EBV proteins. These T cells produced rapid tumor clearance in a xenograft model.

## 4. Conclusions

Due to the COVID-19 outbreak, this symposium was the first hybrid (online and face-to-face) meeting in the history of EBV meetings. The benefits of online meetings other than infection prevention are the lack of travel expenses, low attendance costs, and saving travel time. The chairs and presenters from the United States, Europe, and Asia successfully presented their results and discussed them online in this symposium. One drawback of the online meeting that we were confronted with during its preparation was organizing it while considering the geographic location of each speaker. Because of this issue, the categorization of each session was challenging, and we regret that the presentation style was not as the presenters wished. This meeting provided an overview of exciting topics in the EBV research field and related diseases. The development of modern technologies, for example, CRISPR-based gene knockout and single-cell analysis, enables us to conduct detailed, in-depth, and accurate research on EBV. Hence, we learned that EBV might play various roles in the tumorigenic transformation of diverse immune and epithelial cells. The geographical disparities in EBV-related diseases have not been fully elucidated; however, they may be explained by the sequence variation from different EBV strains. A greater understanding of EBV would boost the establishment of a novel virus-specific treatment for EBV-related malignancies. Thus, we eagerly look forward to the next EBV symposium, which will be held in Siena, Italy, in 2022.

## Figures and Tables

**Figure 1 cancers-14-02924-f001:**
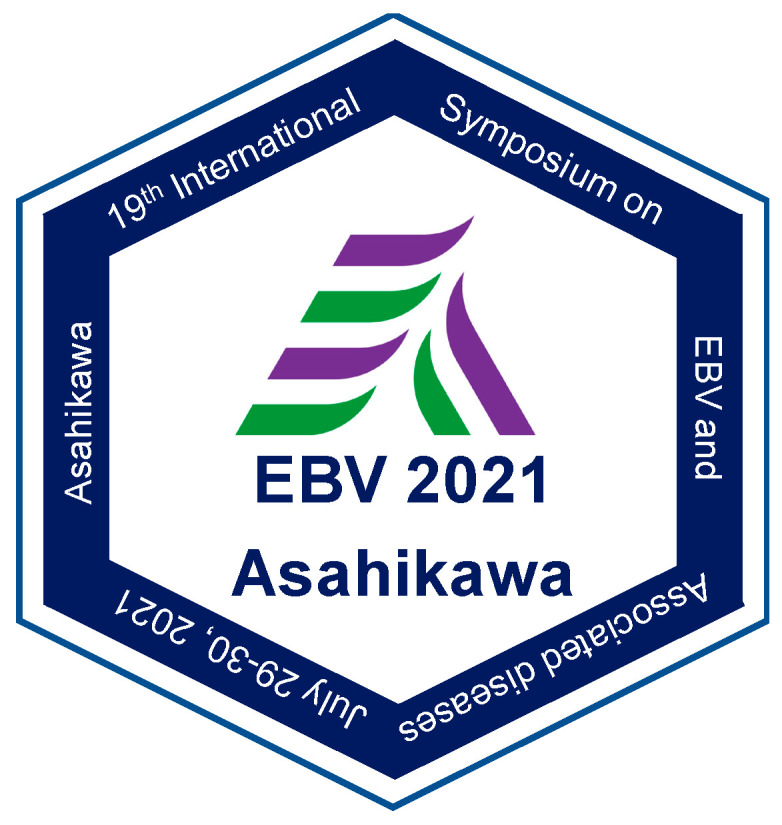
The conference logo of the 19th International Symposium on Epstein–Barr Virus and Associated Diseases.

**Figure 2 cancers-14-02924-f002:**
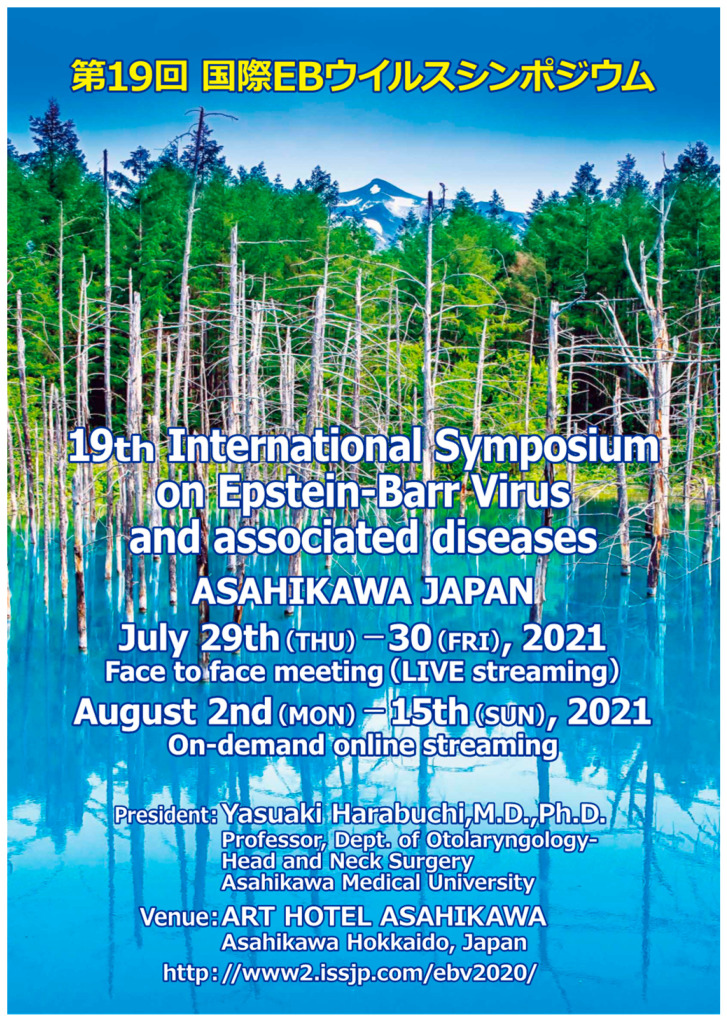
The poster of the 19th International Symposium on Epstein–Barr Virus and Associated Diseases.

