# Peer review of "19th International Symposium on Epstein–Barr Virus and Associated Diseases, 29–30 July 2021, Asahikawa, Japan"

_cancers, 2022, doi:10.3390/cancers14122924_

Round 1

Reviewer 1 Report

Kumai and others report on the 19th International Symposium on Epstein-Barr Virus and Associated Disease that was held July 29-30 in Asahikawa, Japan.  They provide a concise and explicit summary of the major findings presented in the meeting, revealing the remarkable advance in the field of EBV research. EBV is associated with ~200,000 annual new cases of human cancers and is considered responsible for ~145,000 deaths every year.  Therefore, this meeting report should be very informative and interesting to cancer researchers in general. 

This symposium was held as a hybrid (online and face-to-face) meeting and views on the benefits and drawbacks of online meetings from the standpoint of organizers are included in this report.  This part may be particularly useful to those who are planning to organize a meeting of similar type.

Minor points

1. Line 115; Why are three names of authors indicated for this particular presentation, while only one name is indicated for all others?

2. Line 336; The sentence “A genome-wide association study did not detect any EBV genomes in patients with EBV mono-infection” does not seem to make sense.  Could the authors mean that a genome-wide association study did not detect any infectious mono-specific genetic marker in the EBV genome?

Author Response

Reviewer 1:

We thank this reviewer for taking the time for reviewing our manuscript.

Specific Issues:

  1. Line 115; Why are three names of authors indicated for this particular presentation, while only one name is indicated for all others?

Answer: Each of the authors (Atsushi Okabe and Harue Mizokami, both from Astushi Kaneda’s Lab) presented separately in the meeting. Because the subject of their presentations was similar, we decided to summarize their findings in one sentence. We hope the reviewer accept this compilation.

  1. Line 336; The sentence “A genome-wide association study did not detect any EBV genomes in patients with EBV mono-infection” does not seem to make sense. Could the authors mean that a genome-wide association study did not detect any infectious mono-specific genetic marker in the EBV genome?

Answer: We apologize for the confusion. A genome-wide association study did not detect any EBV variant that is specific to EBV mono-infection. We have rephrased the sentence in the revised manuscript.

Reviewer 2 Report

The conference report has comprehensive summarized the update findings presented in the 19th International Symposium on Epstein-Barr Virus and Associated Diseases that held in Asahikawa, Japan from July 29–30, 2021.  A number of important discoveries in this important field have been introduced. It provides a update overview of current understanding of various EBV-associated cancers, especially the NK/T-cell lymphoma, gastric cancer and nasopharyngeal carcinoma. Few sentences in the manuscript are needed to clarify:

-       Lane 124-125: “the natural killer-kappaB (NK-kB) pathway in EBVaGC”. Is it means "NF-kB” pathway ??? nuclear factor-kappaB??

-        Lane 136-137: whether the sentence means: “the mutational pattern of the genome including a new homologous recombination defect signature prevalent in NPC”.

-         LANE 137-138: the sentence should be: “recurrent structural alterations and homozygous deletions alter NF-κB and transforming growth factor (TGF)-β signalling, cell cycle, and impaired immunity.

Author Response

Reviewer 2: We thank this reviewer for taking the time for reviewing our manuscript.

  1. Lane 124-125: “the natural killer-kappaB (NK-kB) pathway in EBVaGC”. Is it means "NF-kB” pathway ??? nuclear factor-kappaB??

Answer: We truly apologize for the typo. This should be nuclear factor-kappaB (NF-kB) as indicated by the reviewer. We have collected this issue in the revised manuscript.

  1. Lane 136-137: whether the sentence means: “the mutational pattern of the genome including a new homologous recombination defect signature prevalent in NPC”.

Answer: We appreciate the reviewer’s correction. This rephrased sentence clearly explains our interpretation of this presentation. We thank the reviewer, and rephrased the sentence as advised.

  1. LANE 137-138: the sentence should be: “recurrent structural alterations and homozygous deletions alter NF-κB and transforming growth factor (TGF)-β signaling, cell cycle, and impaired immunity.

Answer: We have corrected the sentence as advised in the revised manuscript.